# The Role of Hydrogen Sulfide in Plaque Stability

**DOI:** 10.3390/antiox11122356

**Published:** 2022-11-28

**Authors:** Qian Lin, Bin Geng

**Affiliations:** State Key Laboratory of Cardiovascular Disease, National Center for Cardiovascular Diseases, Fuwai Hospital of Chinese Academy of Medical Sciences, Peking Union Medical College, Beijing 100023, China

**Keywords:** atherosclerosis, plaque stability, hydrogen sulfide, cystathionine gamma lyase

## Abstract

Atherosclerosis is the greatest contributor to cardiovascular events and is involved in the majority of deaths worldwide. Plaque rapture or erosion precipitates life-threatening thrombi, resulting in the obstruction blood flow to the heart (acute coronary syndrome), brain (ischemic stroke) or low extremities (peripheral vascular diseases). Among these events, major causation dues to the plaque rupture. Although the initiation, procession, and precise time of controlling plaque rupture are unclear, foam cell formation and apoptosis, cell death, extracellular matrix components, protease expression and activity, local inflammation, intraplaque hemorrhage, and calcification contribute to the plaque instability. These alterations tightly associate with the function regulation of intraplaque various cell populations. Hydrogen sulfide (H_2_S) is gasotransmitter derived from methionine metabolism and exerts a protective role in the genesis of atherosclerosis. Recent progress also showed H_2_S mediated the plaque stability. In this review, we discuss the progress of endogenous H_2_S modulation on functions of vascular smooth muscle cells, monocytes/macrophages, and T cells, and the molecular mechanism in plaque stability.

## 1. Introduction

Hydrogen sulfide (H_2_S) is an identified and recognized gasotransmitter after nitric oxide and carbon oxide. As endogenous methionine catalysis production, H_2_S major generates from homocysteine trans-sulfide metabolism. Cystathionine β synthase (CBS), cystathionine γ lyase (CSE), cysteine aminotransferase (CAT), and 3-mercaptopyruvate sulfur transferase (3-MST) are major synthetases of H_2_S. In cardiovascular tissues, CSE and 3-MST are expressed, and CBS to a much lesser extent. There is no accurate experimental evidence of dominant CSE protein expression in cardiovascular tissues. However, in mouse liver (most endogenous H_2_S generation organ), CSE protein exceeds CBS protein by about 60-fold [1], and CSE mRNA is also about 4.8-fold comparison to CBS and 48.9-fold to 3-MST [2]. Therefore, H_2_S biosynthetic activity might be sourced from CSE. More and more studies highlight that CSE/H_2_S protects against cardiovascular diseases, including atherosclerosis in particular.

Atherosclerosis is one of most prevalent cardiovascular diseases worldwide, characterized as chronic inflammation and lipid accumulation in the large arteries. Atherosclerotic plaque rupture or erosion causes the formation of thrombosis or blood clot, resulting in acute events, such as myocardial infraction, or ischemic stroke, contributing to a major mortality rate in the case of human diseases [3]. Although acute coronary syndrome (ACS) or sudden deaths in younger individuals or women patients are usually triggered by plaque erosion, even superficial erosion [4,5], most cardiovascular events occur due to the plaque rupture. Therefore, plaque stability is crucial for preventing cardiovascular events. Some risk factors (such as hyperhomocysteinemia, hypercholesteremia etc.) promote necrotic core formation due to efferocytosis impairment (causing a reduction of apoptotic foam cells clearance), collagen synthesis decrease, and protease (e.g., matrix metallopeptidase) expression/activity elevation, triggering the thickness of fibrotic cap, local acute or chronic inflammation response, intraplaque hemorrhage, and plaque calcification [6], contributing to the pathophysiological modulation of plaque stability.

CSE/H_2_S can inhibit endothelial inflammation, vascular smooth muscle cell (VSMC) phenotype switch, macrophage polarization and lipid-phage, foam cell formation, and atherothrombosis response, resulting in the blocking atherogenesis [7,8,9]. Recent works have also shown that the treatment of H_2_S donors enhanced plaque stability in a mouse model of ApoE knockout mice [10]. Here, we discuss how CSE/H_2_S modulate atherosclerotic plaque stability, its mechanism, and potential translational merit.

## 2. Intraplaque Cell Types and Interactions in Plaque Stability

Plaque stability is mediated by many complicated factors. In 1989, Muller JE and colleagues encapsulated a hypothetic concept of “vulnerable atherosclerotic plaque”, characterized as collagen exposure, or large necrotic core, based on studies of autopsy and angiographic data from myocardial infraction or sudden cardiac death [11]. In 2000, Virmani R. groups showed a “thin” fibrous cap (<65 μm) plaque was “vulnerable” [12]. However, some in vivo human image clinical trials demonstrated that “vulnerable plaque” did not inevitably rupture or cause thrombotic events [13,14,15]. Although these studies are contradictory, they still imply some “vulnerable” characteristics: such as large necrotic core, more macrophages, VSMC apoptosis and necrosis, reduced the collagen level and active inflammation [16].

In human atherosclerotic plaque, approximately 14 different cell populations were identified, including endothelial cells, VSMCs, myeloid cells, and immunocytes (T cells, B cells and mast cells) [17]. Animal and human evidence has demonstrated that each intraplaque cell population plays a critical role in plaque development and stability.

### 2.1. Endothelial Cells

Endothelial cells (ECs) attach to internal elastic lamina as single layer cells with tight junctions to block blood and vascular wall. Disturbed blood flow, elevated low-density lipoprotein (LDL), heightened blood glucose, or other cytokines and biochemical molecules, cause endothelial proinflammatory activation, which triggers the expression of P-selectin, E-selectin, vascular cellular adhesion molecule-1 (VCAM-1), and intercellular adhesion molecule-1 (ICAM-1) expression, then promote the adhesion of monocytes, macrophages, or other leukocytes [18]. The disturbed blood flow, as well as other factors, also trigger endothelial apoptosis and loss of tight junctions, which aggravate LDL and triglyceride-rich lipoprotein retention in the arterial wall, either by paracellular leak or transcytosis [19]. The lipid retention and inflammation promote early plaque formation and development. Endothelial apoptosis or local inflammation also aggravate plaque erosion or superficial erosion, a procession of neutrophil recruitment, activation, and releasing of neutrophil extracellular trap (NET), promoting endothelial dysfunction and thrombosis formation [6].

### 2.2. Vascular Smooth Muscle Cells

VSMCs are dominant cells in the arterial media and responsible for vascular tone challenging and the secretion of extracellular matrix (ECM), which plays essential roles in vascular compliance and elastic recoil of artery in response to hemodynamics changes. In atherosclerotic plaques, VSMCs contribute to approximately 30–70% all plaque cells according to lineage-tracing studies [20]. Most α-SMA positive cells in the fibrotic cap expressed stem cell antigen 1 (SCA1; also known as LY6A), a marker of mesenchymal stem cells, suggesting they are derived from bone-marrow [21]. Other studies also demonstrate that intracellular VSMCs are generated by the local recruitment and proliferation of few cells, but not derived from arterial medial cells [22,23]. Although plaque VSMCs are generated from different cells, they play an essential regulatory role in all stages of atherosclerosis and numerous processes throughout this disease [20].

In response to damage, VSMCs can transform the contract phenotype into the synthetic phenotype; namely, heighten proliferation and migration-related genes expression, but reduce contraction genes expression. Local VSMCs proliferation with synthetic phenotype characteristics contributes to diffuse intimal thickening (DIT) or intimal xanthomas (fatty streaks), which are considered pre-atherosclerotic plaque [24,25]. In early atherosclerosis, pathological intimal thickening (PIT) is characterized as intimal lipid deposition, with abundant VSMCs and ECM. In this stage, VSMCs contribute to most ECM components; differentiate into phage-like phenotype to form foam cells; promote inflammation and the intimal recruitment of monocytes; and induce microcalcification formation in lipid pool [26,27,28,29]. In the late stage, plaque VSMCs were previously thought to have a beneficial role, because most collagens of fibrous cap formation derived from α-SMA positive cells. In addition, approximately 30–70% foam cells in mouse plaque, 30–40% of CD68^+^ cells, and 50% foam cells in human plaque derived from VSMCs [30,31,32]. Over-inflammation or lipid over-load, aggravate VMSCs death, including apoptosis, autophagic cell death, necrosis, necroptosis, pyroptosis, and paraptosis, resulting in necrotic core formation [33]. VSMC senescence in human and mouse plaque has been widely accepted, which triggers destabilization of atherosclerotic plaques by reducing VSMC numbers and fibrous cap collagens content (due to loss of proliferation potential of senescence cells), driving intraplaque inflammation and ECM degradation [20,33]. Plaque VSMCs can also differentiate into osteochondrogenic cells [16], which promote intraplaque microclacification or spotty calcification, heightening the risk of intraplaque hemorrhage and plaque rupture [34,35]. Collectively, VSMCs play critical role in atherogenesis, plaque rupture.

### 2.3. Monocytes and Macrophages

Substantial clinical and basic studies demonstrated that myeloid cells, including monocytes and macrophages, exert function diversity in the pathogenesis of atherosclerosis and cardiovascular events [28]. In health condition, resident macrophages (mostly derived from circulating monocytes, renew by local proliferation) populate in the adventitia as an innate immune defense [36]. Disturbed blood flow, high level of ox-LDL, or other molecules, such as blood glucose, led to a dysfunction of endothelial cells and impairment of tight junctions, resulting in lipid transendothelial transport into intima, which is then taken up by intima resident macrophages to form foam cells [36,37]. Impaired endothelial cells express adhesion factors, which promote circulatory monocytes adhesion, increasing plaque macrophages accumulation [38]. Therefore, the plaque macrophages represent an essential cellular resource of foam cells.

Macrophages can activate into pro-inflammation M1 polarization state by lipopolysaccharide (LPS)/interferon γ (IFN-γ) or anti-inflammation M2 polarization state by interleukin-4 (IL-4) and IL-13 [39]. Recent single cell sequencing data revealed the co-existence of several macrophage populations, performing pro-inflammation or anti-inflammation functions in the atherosclerotic plaques [40,41]. Another macrophages population expressed canonical M1 (Cd11c, Ccl2 and Il-1β) or M2 (Mrc1, Clec10a and Mgl2) markers. Surprisingly, there are more expression M2-marker resident macrophage populations in the plaque in comparison to M1-marker populations, maybe due to low IL-4 and IL-13 levels in the plaque [42], but not associated with IFN-γ and its receptor [43]. This phenomenon is very interesting, because anti-inflammation therapy, such as CANTOS clinical trial, using anti-IL-1β antibody-canakinumab, significantly reduced recurrent cardiovascular events independent of lipid-lowering [44]. Therefore, macrophages activation and polarization mediate plaque chronic inflammation contribution to plaque stability.

Efferocytosis is a complicated regulation process with many factors involved. Briefly the four steps include: “Find me” (regulatory factors: ApoE4, Cd95l, Cx3cl etc.), “Eat me” (regulatory factors: Sr-b1, Tim-1/Tim-4, Lrp1 etc.), “Engulfment” (regulatory factors: Abca7, Irf5, Cdkn2b, Tlr3, Ppar-γ etc.), and “Releasing and Anti” (Releasing anti-inflammation factors) [45]. Efferocytosis clears intraplaque apoptotic foam cells. Thereby, ROS production and proinflammation mediator are reduced by foam cells, heightening plaque stability [45,46,47]. On the other hand, elimination foam cells also increased anti-oxidation and anti-inflammation production, such as transforming growth factor β (TGF-β), IL-10, heme oxgenase-1 (HO-1) [45]. In a summary, efferocytosis of macrophages attenuates necrotic core formation and intraplaque chronic inflammation [48], promoting the plaque stability.

### 2.4. Adaptive Immune Cells

In human plaques, many adaptive immune cell populations have been identified by single cell sequencing, including T cells, B cells, dendritic cells, and NK cells. Among them, the percentage of T cells is beyond 60% [49]. CD8+ T cells are dominant populations [49]. Cytokines from CD8+ T cells, such as IFN-γ, perforin, and granzyme B, promote plaque inflammation and lesion growth via the lysis of apoptotic cells [50,51] or inducing monopoiesis [52]. More interestingly, intraplaque CD4+ T cells number was heightened in patients with symptomatic disease (recent stroke or transient ischemic attack) compared to asymptomatic disease (no recent stroke) [49], suggesting that CD4+ T cell subsets play a critical role in plaque stability [53]. In a different condition, naïve CD4+ T cells can differentiate into T helper 1 (Th1), Th2, Th17, and regulatory T (Treg) subsets. Th1 cells express and secrete IFN-γ and TNF, exacerbating plaque instability. Th17 cells secrete IL-17 and promote plaque inflammation but enhance plaque stability. Th2 cells resist plaque growth and instability by secreting IL-4, IL-5, and IL-13. Treg cells are majorly enriched in regressing plaques, which not only inhibit local inflammation, but also improve tissue repairment by efferocytosis [53,54].

Until now, the molecular mechanism of plaque rupture/erosion has remained unclear. The present evidence supports that the necrotic core formation, plaque calcification, plaque collagen synthesis or degradation, and local inflammation heighten the risk. A number of studies also highlight that inflammation is a ubiquitous and crucial pathological process in plaque initiation, growth, and rupture/erosion.

## 3. Cystathionine γ Lyase/Hydrogen Sulfide Role in Atherosclerotic Plaque Stability

As a gasotransmitter, H_2_S can easily penetrate the biological membranes due to its lipophilic characteristics. In the body fluid, approximately 1/2 H_2_S remains undissociated form, and 2/3 exists as HS- ion form at equilibrium with H_2_S. Recent studies also show that most H_2_S exists in the body as protein binding form [55]. Thus, binding H_2_S, HS-, and free H_2_S maintain a dynamic equilibrium in the body. Once a damage stimulus is received, such as inflammation or reactive oxygen species (ROS), the metabolic activity is increased in cells, and the pH values of local microenvironment are reduced, which can accelerate free H_2_S generation from HS-. Due to the reduction feature of H_2_S, the constant consumption caused a decrease of local H_2_S level, which led to up-regulation of CSE/H_2_S system (such as acute inflammation [56]) at compensatory phase, or down-regulation of CSE/H_2_S system at uncompensated period. This may be an interpretation of CSE/H_2_S reduction in many cardiovascular diseases [56].

More and more research has shown that CSE/H_2_S play a protective role in atherogenesis [57]. In human and mouse atherosclerotic plaque, CSE protein expression was downregulated [7,58]. Global CSE knockout exacerbated plaque growth by promoting inflammation and oxidative stress [59]. Oppositely, H_2_S donors attenuated atherogenesis by inhibiting endothelial inflammation and foam cell formation [7,8,9]. For the mechanism, CSE/H_2_S may modulate VSMC phenotype switch, macrophage polarization, and lipid-phage, endothelia inflammation and monocytes adhesion, and atherothrombosis response, attributed to the genesis and development of plaque [57,60]. Recently, Xiong Q and colleagues showed that H_2_S enhanced plaque stability by increasing fibrous cap thickness and collagen content, reducing plaque VSMC apoptosis and expression of the collagen-degrading enzyme matrix metallopeptidase-9 (MMP-9) in ApoE knockout mouse model [10]. Here, we discuss the effect of endogenous CSE/H_2_S in plaque stability, regarding essential risk factors and intraplaque cell population functions.

### 3.1. H_2_S Reduced Hypercholesterolemia

Heightening blood cholesterol is a critical process for atherosclerosis initiation and growth. Lowing lipid therapy is the still first-line atherosclerosis therapeutic selection, and it effectively reduces the morbidity and mortality of cardiovascular events. In healthy adult volunteers, circulatory H_2_S level positively correlated with blood HDL-cholesterol level (r = 0.49), but negatively correlated with LDL/HDL ratio (r = −0.39) [61]. Our previous study also showed that H_2_S donors lowered serum total cholesterol and LDL cholesterol [9], in part by increasing LDL receptor (LDLR) expression and enhancing LDL uptake activity [62]. In this mechanism, firstly, H_2_S sulfhydrates Kelch-like ECH-associated protein 1 (Keap-1) at Cys151 site, then promotes nuclear erythroid 2-related factor 2 (Nrf-2) dissociation and nuclear translocation, and up-regulates LDL-related protein 1 (LRP-1) expression in hepatocytes [63]. Secondly, H_2_S activates the phosphoinositide 3-kinase (PI3K)/protein kinase B (Akt)-sterol regulatory element binding proteins 2 (SREBP-2) signaling pathway and inhibits proprotein convertase subtilisin/kexin type 9 (PCSK9), causing LDLR protein elevation [62], which is not dependent on liver X receptor α (LXRα) [64].

### 3.2. H_2_S Attenuates Hyperhomocysteinemia

Hyperhomocysteinemia is an independent risk factor for atherosclerosis. Some studies also supported that high blood total homocysteine (Hcy) level correlated with cardiovascular events. However, lowering homocysteine therapy by B type vitamin and/or folic acid by enhancing Hcy re-methylation do not effectively reduce morbidity and mortality of cardiovascular events, but just slightly attenuate stroke mortality for those presenting with hypertension and hyperhomocysteinemia [65]. H_2_S is a product of homocysteine trans-sulfur catalysis. H_2_S can increase CSE catalysis Hcy activity or methylenetetrahydrofolate reductase activity by sulfhydration, resulting in the reduction of serum Hcy, then attenuate Hcy-induced macrophages infiltration in the plaque and plaque area [66,67]. In vitro, H_2_S also reduced Hcy-induced reactive oxygen species (ROS) and endoplasmic reticulum stress [68,69]. These studies also provide a new therapeutic clue using H_2_S donors, in order to prevent cardiovascular events in patients suffering from hyperhomocysteinemia.

### 3.3. H_2_S Modulates VSMC Function in Pathogenesis of Plaque Stability

VSMC function contributes to the formation, growth/development, and even rupture of atherosclerotic plaque [20]. In the arterial wall, endogenous H_2_S dominant derived from VSMCs [70]. Numerous evidence, direct and indirect, including in vivo and in vitro studies, demonstrated that the endogenous CSE/H_2_S system may mediate VSMC proliferation, cell death (apoptosis, autophagy, ferroptosis), phage-like differentiation, and calcification, to participate in the pathophysiological process of plaque stability (Figure 1).

#### 3.3.1. VSMC Proliferation

In vitro, H_2_S can inhibit VSMC proliferation induced by serum, endothelin-1 [71], and angiotensin II [72]. The MAPK activation, Foxo1 sulfhydration, mitochondrial dynamics (fission and fusion), pyruvate dehydrogenase E1 subunit alpha 1 (Pdha1), and brahma-related gene 1 (the central catalytic subunit of the SWI/SNF apparatus (an ATP-dependent chromatin remodeling complex) are involved in the regulatory mechanism [71,72,73,74,75]. CSE deficient VSMCs exhibit high cell numbers and Brdu incorporation ability by increasing ERK1/2 phosphorylation and reducing cyclin D1 protein expression. In global CSE knockout mouse aorta, calcitonin receptor-like, integrin beta 1, and heparin-binding epidermal growth factor-like growth factor transcription expression also increased compared with that of the wild type mouse [76]. The VSMC proliferation inhibition by H_2_S limits arterial hyperplasia and remodeling [72,77], reducing early plaque development.

#### 3.3.2. VSMC Phage-like Differentiation and Apoptosis

In human or mouse plaques, approximately 30–70% foam cells express both macrophage marker (CD68) and VSMC marker (α-SMA), indicating that many local proliferation VSMCs differentiated into phage-like phenotype to form foam cells. Oxidative-low density lipoprotein cholesterol (ox-LDL) can induce VMSC phage-like differentiation, characterized as intracellular lipid deposition and expression of CD68. Our recent study highlights that VSMC endogenous CSE expression declines in vulnerable plaques compared to stable plaques of patients. More intriguingly, the relative CSE expression quantification in α-SMA positive cells was negatively correlated with plaque vulnerable index with an R value of approximately –0.9 [58]. The overexpression CSE or H_2_S donor attenuated ox-LDL-stimulated VMSC phage-like differentiation [58]. Intraplaque persistent inflammation, or ROS accumulation, may cause the apoptosis of phage-like VSMCs; if they are not effectively removed, the necrotic core is susceptible to form and expand. H_2_S donors can reduce ox-LDL-triggered VSMC caspase-3/9 activity, Bax/Bcl-2 ratio, and Lox-1 mRNA expression; reduce lipid oxidation [78], resulting in attenuation of VSMC apoptosis [10]. H_2_S also inhibits ox-LDL induced iNOS expression, enhancing protein S-nitrosylation to block VSMC apoptosis [79]. Thus, VSMC endogenous CSE/H_2_S inhibits VSMC inflammation and ROS generation, and reduces phage-like differentiation and apoptosis, which effectively attenuates necrotic core formation/expansion, to elevate the plaque stability.

#### 3.3.3. VSMC Ferroptosis

Ferroptosis is a novel cell death process due to cysteine-depletion and glutamate induced cytotoxicity [80]. During advance plaque formation, lipid oxidation, intraplaque hemorrhages, and ion deposition might promote local ferroptosis, which is also positively correlated with plaque stability. The overexpression of glutathione peroxidase 4 (Gpx4) (reducing ferroptosis), or treatment with ferrostatin-1 (ferroptosis inhibitor), significantly attenuated the development of plaque [81,82]. In human plaques, prostaglandin-endoperoxide synthase 2 (Ptgs2) and anti-acyl-CoA synthetase long-chain family member 4 (Acsl4) expression were up-regulated, and GPX4 protein expression was down-regulated. The plaque Ptgs2 and Acsl4 were positively, but Gpx4 negatively, correlated with the severity of atherosclerosis, indicating pyroptosis and ferroptosis association with plaque stability [83]. Our work demonstrated that CSE/H_2_S depletion exacerbated VSMC ferroptosis association with GSH level decline and Gpx4 expression down-regulation. By contrast, H_2_S donor treatment reduced RLS-3 induced VSMC ferroptosis [84]. Therefore, CSE/H_2_S might modulate VSMC ferroptosis, contribution to the plaque stability.

#### 3.3.4. VSMC Autophagy

Autophagy is an intracellular adaptive response for the lysosome-mediated degradation of damaged cytosolic species, to regulate cell renovation and homeostasis. VSMC specific deletion human antigen R (HuR) or autophagy related 7 (ATG7) [85,86] caused autophagy defective, which promoted VSMC senescence and death, resulting in plaque instability. So, VSMC autophagy exerts a protective role in plaque stability [87].

CSE/H_2_S system can enhance VSMC autophagy during the genesis of plaque. Ox-LDL attenuated VSMC autophagy related gene expression, e.g., LC3, ATG5, BCN1, LAMP1, LC3II/LC3I ratio, and autophagy flux. The deletion of CSE aggravated ox-LDL-induced VSMC autophagy decline. Oppositely, the overexpression of CSE reversed it [58]. In the mechanism, CSE/H_2_S can sulfhydrate transcript factor EB (a mast transcript factor for autophagy and lysosome biogenesis) at cysteine 212 site, heighten its transcription activity [58]. Therefore, CSE/H_2_S can regulate VSMC autophagy to mediate VSMC-phagy-like differentiation and foam cell formation, proliferation, migration, and death, then preserve plaque stability.

#### 3.3.5. VSMC Calcification

Calcification and microcalcification are often observed in the necrotic core or the surrounding ECM, which is an active interrelated process, including macrophage-derived and VSMC-derived calcifying microvesicles, apoptotic bodies releasing, or VSMC osteochondrogenic differentiation [20]. Apoptotic bodies, oxLDL, and pro-inflammatory cytokines promote VSMC osteochondrogenic conversion [88,89,90,91], partly by activation runt-related transcription factor 2 (Runx2) [92]. In vivo mediating VSMC osteochondrogenesis also alters atherosclerotic plaque calcification area [93,94,95]. The local calcification or microcalcification can form calcium nodules, which increase intraplaque hemorrhage risk, or protrude into vessel lumen and precipitate thrombosis, resulting in plaque instable [20].

CSE/H_2_S can regulate Runx2 protein expression [96] and its activity by sulfhydration modification [97] to inhibit VSMC osteochondrogenic differentiation, reducing vascular calcification [98,99]. Furthermore, H_2_S also inhibits Keap1, enhances Nrf2 nuclear activity to heighten NAD(P)H dehydrogenase [quinone] 1 expression [100], or sulfhydrates Cys259 residue in Stat3, inhibiting its activity and cathepsin S [101], resulting in VSMC calcification attenuation. Therefore, CSE/H_2_S can mediate VMSC osteochondrogenic differentiation and inhibit plaque calcification/microcalcification to enhance plaque stability.

#### 3.3.6. VSMC Collagen Synthesis and Extracellular Matrix Secretion

The stable atherosclerosis plaque lumen surface covers a layer of fibrous cap, which composes of collagen, elastin, and populated by a number of α-SMA+ myofibroblast-like cells, macrophages, and lymphocytes. Most α-SMA+ myofibroblast-like cells (60–80%) derived from VSMCs, and others from endothelial-mesenchymal transition (EMT) and macrophage-mesenchymal transition (MMT) [102]. Although EMT and MMT may in the short-term compensate for the composition of fibrous cap, the mice with VSMC special deletion pdgfrb increased indices of plaque instability while prolong the time of high fat diet for 25 weeks [102]. Deletion collagen type XV 1 (Col15a1) in VSMCs significantly reduced collagen content of plaque (>70%), suggesting that most of collagen in the plaques were derived from VSMCs, and which also confirmed by other studies involving different inducers [103,104,105]. These studies indicate VMSC-derived fibrous cap is essential for plaque stability.

In cultured VSMCs, ox-LDL-induced collagen I expression in CSE deficient VSMCs were lower than that of wild type; by contrast, overexpression CSE or H_2_S donor treatment increased collagen I expression [58]. Except for collagen expression, its degradation related enzymes: MMPs and its endogenous inhibitors (tissue inhibitors of MMPs, TIMP) also contribute to fibrous cap formation [27]. Under TNFα stimulation, CSE deficient VSMCs increased MMP9 and MMP2 protein expression and their enzyme activity; on the contrary, H_2_S donor-NaHS reduced them [106]. In cerebrovascular cells, homocysteine induced MMP13 elevation but TIMP-1 reduction, whereas, overexpressed CBS, CSE and 3-MST normalized their expression [107]. In a carotid artery injury mouse model, NaHS treatment also attenuated MMP9 expression in VSMCs [108]. In the molecular mechanism, CSE/H_2_S S-sulfhydrated SP1, blocking its transcript activity, then increasing MMP2 gene transcription. CSE/H_2_S also induced MMPs per se S-sulfhydration at cysteine sites, inhibiting their binding to zinc ion then lowering their enzyme activity [106]. These in vivo and in vitro studies demonstrate that CSE/H_2_S promotes VSMC collagen synthesis and inhibits MMPs to reduce collagen degradation, resulting in stabilization of the plaque.

### 3.4. H_2_S Targets Macrophage Polarization, Foam Cell Formation and Efferocytosis

In the early atherosclerosis stage, local endothelial inflammation promotes monocytes adhesion, then migration into the intimal space. The intimal monocytes proliferate and differentiate into macrophage in initial of atherosclerotic plaque. Plaque macrophages respond to local inflammation and exacerbate the inflammatory cycle by producing proinflammatory cytokines and ECM components, then heightening the retention of lipoprotein to form foam cells [28]. Plaque accumulated cytokines also induce macrophages activation to polarize into pro-inflammation M1 type or anti-inflammation M2 type, promoting (M1) or regressing (M2) plaque growth. Chronic inflammation can promote foam cells apoptosis, lower the efferocytosis of macrophage, resulting in promoting the necrotic core formation and plaque instability [39]. Macrophages also express the CSE/H_2_S system, which mediates the macrophage polarization, lipid-phage, and efferocytosis to maintain the plaque stability (Figure 2).

#### 3.4.1. H_2_S Mediates Macrophage Polarization

Macrophage polarization is a dynamic process. M1 macrophages polarize in response to lipopolysaccharides (LPS), lipoproteins, toll-like receptor ligands, interferons, and pathogen-associated molecular complexes [39]. M1 can produce pro-inflammatory factors, such as IL-1β, IL-6, TNF-α, and some protease to destruct local tissues. Of course, nuclear factor kappa B (NF-κB) and signal transducer and activator of transcription-1 (Stat-1) signaling contribute to M1 polarization, in glycolysis as fuels [38,39]. M2 macrophages are polarized in response to IL-4 and IL-13, exhibit anti-inflammation properties by secretion of IL-10 and IL-1 receptor agonist, and stabilize plaque, with fatty acid oxidation as energy metabolic properties [38,39]. In human plaques, M1 macrophage markers were expressed in rupture-prone and unstable area, but M2 macrophage localized in a stable area [109]. The induction of M2 polarization or giving M2 macrophage-derived anti-inflammatory factors attenuate atherosclerosis development [39].

Canonical M1 macrophage inducer-LPS heightened CSE expression, which inhibited jumonji domain-containing protein 3 (JMJD3) demethylase expression, thereby increased H3K27me3 to lower inflammation-related genes expression [110]. On the contrary, H_2_S donor inhibited LPS-induced pro-inflammatory cytokines, such as IL-6, TNF-α [111], IL-1β, and PEG2 [112], and histone H3 acetylation at the promoter of these cytokines attributed to the modulation [111]. H_2_S donors also abrogate M1 polarization [113] by reducing NOX4 derived ROS generation [114], or inhibiting NF-κB activity by S-sulfhydation P65 subunit at Cys38 site [56,115]. Sirtuin1 is a class III histone deacetylase, can deacetylate P65 subunit and histone H3, then inhibit cellular inflammation. H_2_S also sulfhydrates sirtuin1 at zinc ion binding domain to enhance its activity [9], which might contribute to the molecular mechanism of M1 polarization.

By contrast, H_2_S and its endogenous generation enzymes (CSE and CBS) can promote macrophages M2 polarization. In myocardial infraction (MI) mouse model, H_2_S donor elevated M2 macrophages infiltration in the MI area. H_2_S donor-induced M2 polarization also promote would healing [116]. In the nervous system, a NO-H_2_S double donor-NOSH-NBP heightens microglial/macrophage switch from M1 to M2 polarization, then attenuates ischemia induced neuroinflammation [117]. H_2_S also promotes microglial M2 polarization but inhibits M1 polarization in LPS-induced neuroinflammation model [118]. In the mechanism, H_2_S inhibits macrophage M1 polarization by TLR4/MyD88/NF-κB pathway and NLRP3 inflammasome; whereas H_2_S promotes M2 polarization by sulfhydration peroxisome proliferator activated receptor gamma (PPARγ) [117,119], or mediating mitochondrial biogenesis [120]. Although there are no direct data in the model of atherosclerosis, H_2_S-mediated macrophage polarization is involved in the regulation of plaque stability.

#### 3.4.2. H_2_S Mediates Foam Cells Formation

Macrophage-derived foam cell formation is essential in atherosclerotic plaque occurrence and development. Macrophage-derived foam cell formation is tightly correlated with LDL cholesterol uptake (LDL receptor and related regulatory protein and transcription factors modulated) and efflux (ATP-binding cassette transporter A1(ABCA1), ABCG1 and the nuclear receptors-LXR (liver X receptor)-α and LXR-β mediated) [121]. Local inflammation and antioxidant molecules (heme oxygenase 1, thioredoxin reductase 1, and sulfiredoxin-1 etc.) [39] also contribute to the regulation of LDL uptake/efflux. The over-retention of lipid, inflammation, oxidative stress, and endoplasmic reticulum stress also promote foam cell apoptosis, which exacerbate necrotic core formation/expanding, causing plaque instable.

Elevated homocysteine can promote macrophage inflammation and foam cell formation. Homocysteine up-regulates DNA methyltransferase and induces CSE promoter hypermethylation, causing CSE transcription and H_2_S production reducing, therefore increasing pro-inflammation cytokines secretion in macrophages [122]. Ox-LDL also induce the hypermethylation of CSE promoter [123,124], which is dependent on JNK activation and NF-κB signaling [124]. MiR-216a can bind 3′-UTR of CSE mRNA, blocking expression of CSE gene, and this is an essential mechanism for miR-216a, which promotes formation of foam cell [125]. H_2_S donor inhibits ox-LDL-induced CD36, scavenger receptor A (SR-A) and acyl-coenzyme A cholesterol acyltransferase-1 (ACAT-1) expression, thereby lowering cholesterol uptake of macrophages; by contrast, CSE inhibitor heightens them [126,127]. A new H_2_S donor based on nicotinic acid and chlorfibrate structure also inhibits foam cell formation and pro-inflammatory cytokines secretion, and promotes IL-10 expression by PI3K-Akt pathway [128]. CSE/H_2_S also heightens ABCA1 expression, then enhancing macrophage cholesterol efflux [125]. Our recent study also confirmed the CSE/H_2_S effect on formation of foam cell, in part through sulfhydrating sirtin-1 to increase its deacetylation activity, inhibiting macrophage inflammation and cholesterol uptake, but not the efflux of cholesterol [9]. Collectively, macrophage endogenous CSE/H_2_S system decreases cholesterol uptake, increases cholesterol efflux, reduces inflammation and oxidative stress [129], thereby blocking macrophage-derived foam cell formation, contributing to plaque stability.

#### 3.4.3. The Possible Role of H_2_S in Efferocytosis

Efferocytosis, a process involving the removal of apoptotic cells by macrophages or neighboring cells to acquire a macrophage-like phenotype (such as VMSCs in plaque), play a crucial role in plaque genesis and stability [45,47]. Reduction/inhibition of efferocytosis, or up-regulation of “Don’t eat me” signal molecules (such as CD47, CD24), were confirmed in plaques of human or mice, and promoted the enlargement of necrotic core, accumulation of death cells, and inflammation, resulting in plaque instability [47].

Until now, there is no direct experimental evidence of H_2_S regulation on efferocytosis. However, some studies showed the possibility of H_2_S in all efferocytosis processes. In mouse macrophage cell-line, H_2_S donor dose-dependently inhibited LPS and IFN-γ-induced CX3CR1 and CX3CL1 expression, which are a pair of well-known “Find me” signal molecules [130]. H_2_S sulfhydrated Keap1 at Cys151 site, and activated Nrf2, and then heightened the expression of LRP-1 (an essential marker of “Eat me” process) [63]. H_2_S also attenuated hypoxia or ox-LDL-induced inflammation in THP-1 cell line by inhibiting p38 MAPK signaling, which also a crucial signal for “Engulfment” [131,132]. By contrast, CD47 is a famous molecule “Don’t eat me” signal to reduce efferocytosis. Thrombospondin-1, a ligand of CD47, blocked H_2_S-induced T cell activation and inhibited CSE enzyme expression [133]. These studies indicate CSE/H_2_S may enhance efferocytosis by targeting its all processes, but this needs to be more clearly demonstrated in the future.

### 3.5. H_2_S Mediated Adaptive Immunity to Participate into Plaque Stability

Pathological evidence, single-cell RNA sequencing, and cytometry by time of flight (mass cytometry) approaches confirm the presence of T and B cells in human and mouse plaques. In all leukocytes of plaque, approximately 25%–38% are CD3+ cells, and CD4+ T cells approximately 10% [54]. After antigen-presentation, CD4+ T cells differentiated into distinct Th subsets: Th1, Th2, Th17, Treg, T-follicular helper cells (Tfh), and Type 1 regulatory (Tr1) cells [54]. Th1 subset, special expressing T-bet (T-box transcription factor) and secreting IFN-γ, is dominant in human plaques. IFN-γ inhibits VSMC proliferation but increases M1 polarization [134]. Tregs especially express transcription factor FoxP3 (forkhead box P3) and CD25 (IL-2 high affinity receptor), and play a protective role in plaque stability, by releasing IL-10, TGF-β, and inhibiting T-effector cells [135,136]. Therefore, CD4+ T cells regulate local inflammation to affect necrotic core and fibrotic cap formation, contributing to plaque stability.

In primary mouse T lymphocytes and human Jurkat T cells, H_2_S increases CD69, CD25, and IL-2 expression and promotes T cell proliferation, suggesting T cell activation [137]. In contrast, activated T cells up-regulated CBS and CSE proteins, resulting in increasing H_2_S production [137]. Interestingly, thrombospondin-1 (CD47 suppressor) mediated T activation is also H_2_S-dependent [133]. Except for T cell activation, H_2_S also mediates Tregs proliferation and differentiation. Deletion of CBS [138] or CSE [139] reduced TGF-β-induced Tregs differentiation. By contrast, H_2_S donors promote Tregs proliferation and differentiation [139]. In the mechanism, H_2_S sulfhydrated nuclear transcription factor Y subunit beta (NFYB) facilitated methylcytosine dioxygenases Tet1 and Tet2 expression, causing the elevation FoxP3 transcript by its promoter hypomethylation [138]. H_2_S also activates energy-sensitive AMPK by sulfhydrating live kinase B1 (LKB1) to promote Tregs proliferation and differentiation [139]. H_2_S heightens local Tregs numbers and IL-10 release, attenuating vascular inflammation and T cell infiltration [139], which may increase plaque stability.

## 4. Conclusions and Perspective

Within the past 5–10 years, substantial direct or potential evidence support that CSE/H_2_S can reduce macrophage/VSMC-derived foam cells formation, inhibit cell death, heighten efferocytosis and then lower necrotic core, increase collagen synthesis to form a fibrotic cap, and enhance T cell anti-inflammation differentiation, thus increasing plaque stability. However, there are still some important issues of CSE/H_2_S regulation in plaque stability that need to be investigated in the future. In patients and mice, the plaque CSE/H_2_S system is down-regulated, but its spatiotemporal expression in various cells, its relationship with plaque development, and potential regulatory mechanisms are still unknown. Although experimental evidence showed that CSE/H_2_S may maintain the plaque stability [58], there is no direct evidence that endogenous CSE/H_2_S has been associated with acute coronary syndrome. Therefore, genomic modified (conditional knockout or knock-in) animal models should be investigated in the future, as well as large animal (pig or monkey) models to obtain in vivo solid evidence. Some H_2_S donors in a clinical trial: SG10002 in the case of heart failure, ATB-346 in the case of osteoarthritis, and ammonium tetrathiomolybdate in the case of cancer [140], should also be used in animal models, so as to provide a potential therapeutic possibility for the prevention of cardiovascular events.

## Figures and Tables

**Figure 1 antioxidants-11-02356-f001:**
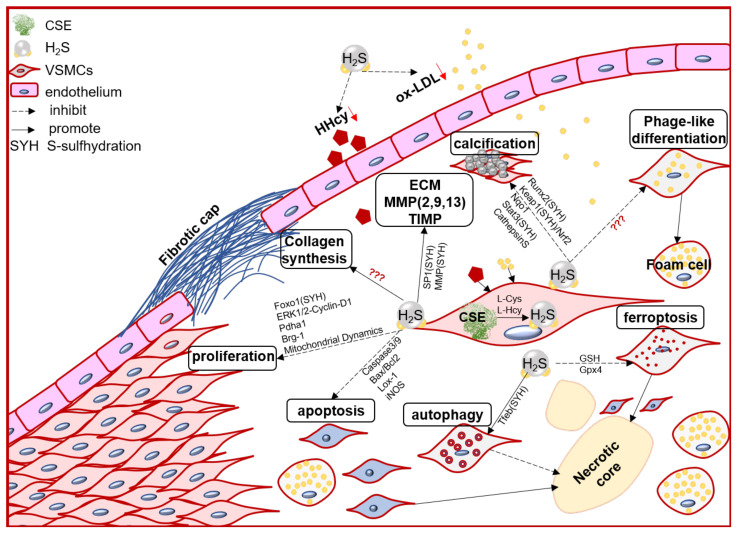
The schematic diagram of endogenous CSE/H_2_S biological modulation and molecular mechanism in VSMC proliferation, apoptosis, ferroptosis, autophagy, phage-like differentiation, osteochondrogenic conversion/calcification, collagen synthesis/secretion, matrix protease expression and activity, engage in the pathogenesis of atherosclerotic plaque and its stability regulation. Foxo1: forkhead box O1; ERK: Extracellular signal-regulated kinase Stat3: signal transducer and activator of transcription 3; Pdha1: pyruvate dehydrogenase E1 alpha 1; Brg-1: brahma-related gene 1; Lox-1: oxidized low density lipoprotein receptor 1; Runx2: runt related transcription factor 2; Nqo1: NAD(P)H quinone dehydrogenase1; Keap1: Kelch-like ECH-associated protein 1; Nrf2: nuclear factor (erythroid-derived 2)-like 2; GSH: glutathione; Gpx4: glutathione peroxidase 4; Tfeb: transcription factor EB.

**Figure 2 antioxidants-11-02356-f002:**
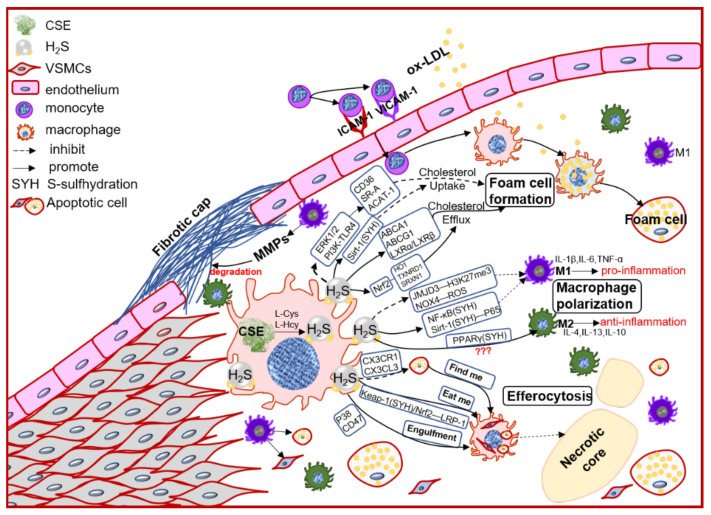
The schematic diagram of CSE/H_2_S modulation in macrophages polarization, macrophage-derived foam cell formation and efferocytosis, and then participation in the plaque stability. PI3K: phosphatidylinositol 3-kinase; TLR4: toll-like receptor 4; Sirt1: sirtuin 1; ABCA1: ATP-binding cassette transporter A1; ABCG1: ATP-binding cassette transporter G1; SR-A: scavenger receptor A; LXR: liver X receptor; HO1: heme oxygenase 1; TXNRD1: thioredoxin reductase 1; SRXN1: sulfiredoxin-1; JMJD3: jumonji domain-containing protein 3; PPARγ: peroxisome proliferator activated receptor gamma; ACAT-1: acyl-coenzyme A cholesterol acyltransferase-1; NOX4: NADPH oxidase 4; ROS: reactive oxidant species; CX3CR1: chemokine (C-X3-C motif) receptor 1; CX3CL1: chemokine (C-X3-C motif) ligand 1; LRP1: LDL receptor related protein 1; IL-1β: interleukin 1 beta; IL-4: interleukin 4; IL-6: interleukin 6; IL-10: interleukin 10; IL-13: interleukin 13; TNFα: tumor necrosis factor alpha.

## Data Availability

Not applicable.

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
