# Peer review of "The Role of Hydrogen Sulfide in Plaque Stability"

_antioxidants, 2022, doi:10.3390/antiox11122356_

Round 1

Reviewer 1 Report

Lin and Geng have compiled a review article focusing on the role of endogenous H2S on atherogenic processes.

 Comments:

The article needs a comprehensive revision. On the one hand, the English language is often used incorrectly and, on the other hand, there are also content-related, subject-specific errors. Just two examples on the first page:

'Atherosclerotic plaque rupture or erosion induces thrombus or blood clot, arising acute events such as myocardial infraction infarction, ischemic stroke or peripheral vascular diseases, and which contributes to the major mortality of human diseases [1].'

-       No, PVD is not an acute event.

'In human atherosclerotic plaque, about 14 different cell populations were identified including endothelium, vascular smooth muscle cells (VSMCs), myeloid cells and immunocytes (T cells, B cells and mast cells)[2], all of them play critical role in plaque development, stability, or erosion.'

-       In principle, the endothelium does not mean a population of cells, but an organ.

Author Response

Lin and Geng have compiled a review article focusing on the role of endogenous H2S on atherogenic processes.

 Comments:

The article needs a comprehensive revision. On the one hand, the English language is often used incorrectly and, on the other hand, there are also content-related, subject-specific errors. Just two examples on the first page:

'Atherosclerotic plaque rupture or erosion induces thrombus or blood clot, arising acute events such as myocardial infraction infarction, ischemic stroke or peripheral vascular diseases, and which contributes to the major mortality of human diseases [1].'

No, PVD is not an acute event.

Rebuttal: We thank the reviewer’s comments. We’ve revised the wording, rearranged the text, and removed PVD from this paragraph, as your suggestion.

'In human atherosclerotic plaque, about 14 different cell populations were identified including endothelium, vascular smooth muscle cells (VSMCs), myeloid cells and immunocytes (T cells, B cells and mast cells)[2], all of them play critical role in plaque development, stability, or erosion.'

In principle, the endothelium does not mean a population of cells, but an organ.

Rebuttal: Very professional comments. We have replaced “endothelium” as “endothelial cells”. Thanks again!

Reviewer 2 Report

I read this review manuscript. I found this manuscript contains informative knowledge. However, I also thought there are other knowledge that also should be described in this review article. 

1. I thought this review describing not only about the plaque stability but also about the developing atherosclerosis. Then the title of this manuscript might inappropriate. 

2. In abstract, this author mentioned about the importance of the targeting plaque stability and erosion in the present atherosclerosis, as a strategy for cardiovascular disease prevention. However, according to description in abstract, this author focusing on the hydrogen sulfide (H2S), especially for its protective role in pathogenesis of atherosclerosis. Then those sentences are not connected each other. The influence of H2S on plaque stability and erosion also should be mentioned.

3. In abstract, brief description about the role of vascular smooth muscle cells, monocytes and macrophages, and T cells on developing atherosclerosis, plaque stability and erosion should be shown.

4. Even the title of this manuscript is “The role of hydrogen sulfide in plaque stability”, there is no description about hydrogen sulfide in the first paragraph of this manuscript. Brief description about the hydrogen sulfide and its relation to plaque is mandatory in the first paragraph of this manuscript.

 5. From present description, the reason why this author interested in “hydrogen sulfide” is not clear. The reason why discussing about hydrogen sulfide is worth in present manuscript should be clarified in the first paragraph of present manuscript.

 6. Biochemical characteristics of hydrogen sulfide also should be discussed. The association between non-dissociative type and dissociative type in relation to pH might also important characteristics when discussing bout hydrogen sulfide. And discussing about the pH levels which associated with atherosclerosis and status of plaque also might be informative. In addition to that, permeability of hydrogen sulfide to lipid bilayer also informative.

 7. How is the influence of differences of hydrogen sulfide type such as free hydrogen sulfide, sulfane sulfur, bound sulfur on present associations?

 8. This author describing about the potential influence of medication on hydrogen sulfide in conclusion section. Then this author also should make a paragraph that explain the potential influence of medication on hydrogen sulfide.

Author Response

Reviewer2

I read this review manuscript. I found this manuscript contains informative knowledge. However, I also thought there are other knowledge that also should be described in this review article. 

  1. I thought this review describing not only about the plaque stability but also about the developing atherosclerosis. Then the title of this manuscript might inappropriate. 

Rebuttal: We thank you for the professional suggestion. Atherosclerosis is a complicated chronic disease. Different pathophysiological modulations exist in different atherosclerotic phase. Here, we were particularly concerned about the possible roles of H2S in the stability of plaques. However, we have to introduce a basic information on the development of the atherosclerosis process, to be able to understand the role of the plaque in the stability of the plaque. Therefore, we didn't change the title because there was too much data. Such as endogenous cells, such cells are crucial to the formation of plaques and the development of plaques, but they don't have a direct correlation with the stability of the plaques.

  1. In abstract, this author mentioned about the importance of the targeting plaque stability and erosion in the present atherosclerosis, as a strategy for cardiovascular disease prevention. However, according to description in abstract, this author focusing on the hydrogen sulfide (H2S), especially for its protective role in pathogenesis of atherosclerosis. Then those sentences are not connected each other. The influence of H2S on plaque stability and erosion also should be mentioned.

Rebuttal: Good comments. We have added some content to link the context. New abstract as following

Abstract: Atherosclerosis is most contribution to cardiovascular events, and involves in the majority deaths worldwide. Plaque rapture or erosion, precipitates life-threatening thrombi, resulting in obstruction blood flow of heart (acute coronary syndrome), brain (ischemic stroke) or low extremities (peripheral vascular diseases). Among these events, major causation dues to the plaque rupture. Although initiation, procession and precise time of controlling plaque rupture are unclear, foam cell’s formation and apoptosis, cell death, extracellular matrix components, protease expression and activity, local inflammation, intraplaque hemorrhage and calcification contribute to the plaque instability. These alterations tightly associate with function regulation of intraplaque various cell populations. Hydrogen sulfide (H2S) is gasotransmitter derived from methionine metabolism, and exerts a protective role in genesis of atherosclerosis. Recent progress also showed H2S mediated the plaque stability. In this review, we discuss the progress of endogenous H2S modulation on functions of vascular smooth muscle cells, monocytes/macrophages, and T cells, and the molecular mechanism in plaque stability.”

  1. In abstract, brief description about the role of vascular smooth muscle cells, monocytes and macrophages, and T cells on developing atherosclerosis, plaque stability and erosion should be shown.

Rebuttal: We added this content (see the previous one rebuttal).

  1. Even the title of this manuscript is “The role of hydrogen sulfide in plaque stability”, there is no description about hydrogen sulfide in the first paragraph of this manuscript. Brief description about the hydrogen sulfide and its relation to plaque is mandatory in the first paragraph of this manuscript.

Rebuttal: An essential comment. We adjusted the context of this revision. We first introduced the H2S modulation for atherosclerosis and plaque stability.

  1. From present description, the reason why this author interested in “hydrogen sulfide” is not clear. The reason why discussing about hydrogen sulfide is worth in present manuscript should be clarified in the first paragraph of present manuscript.

Rebuttal: Yes, we added the content in the first paragraph.

  1. Biochemical characteristics of hydrogen sulfide also should be discussed. The association between non-dissociative type and dissociative type in relation to pH might also important characteristics when discussing bout hydrogen sulfide. And discussing about the pH levels which associated with atherosclerosis and status of plaque also might be informative. In addition to that, permeability of hydrogen sulfide to lipid bilayer also informative.

Rebuttal: As your comment, we added a paragraph (page 4, line 177) to discuss the H2S chemical characteristics, and these characteristics per se regulation on atherosclerosis.

  1. How is the influence of differences of hydrogen sulfide type such as free hydrogen sulfide, sulfane sulfur, bound sulfur on present associations?

Rebuttal: This comment is very important. However, we just discussed that the H2S, HS- and bound sulfur may maintain a dynamic equilibrium in the body fluid. We did not discuss the sulfane sulfur and bound sulfur function due to the compatibility with our aim of this review, and little research data for atherosclerosis.

  1. This author describing about the potential influence of medication on hydrogen sulfide in conclusion section. Then this author also should make a paragraph that explain the potential influence of medication on hydrogen sulfide.

Rebuttal: Thanks a lot. We rewrote the conclusion and perspective, and we added a bit of content for the limitation and possible translational benefit of H2S on the stability of plaques.

Reviewer 3 Report

·      Very interesting review which underlines the role of hydrogen sulfide in the protection against cardiovascular events.

·      Some modifications are required.

·      The parameters that regulate plaque stability are essentially levels of inflammation and calcification. For the latter, it is the microcalcification that seems to significantly influence the stability of the atherosclerotic plaque. The authors have extensively discussed plaque calcification in general, but it would be relevant to discuss microcalcification more specifically and how H2S can regulate it.

·      A paragraph should be added to clarify the potential role of H2S on plaque stability and the prevention of cardiovascular events. Indeed, much of the literature cited in this manuscript concerns animal models and the translation to humans remains to be established. The authors have discussed this point in the conclusion part, but it would be more important to discuss it in a separate paragraph.

Author Response

Reviewer3

Very interesting review which underlines the role of hydrogen sulfide in the protection against cardiovascular events.

  • Some modifications are required.

  • The parameters that regulate plaque stability are essentially levels of inflammation and calcification. For the latter, it is the microcalcification that seems to significantly influence the stability of the atherosclerotic plaque. The authors have extensively discussed plaque calcification in general, but it would be relevant to discuss microcalcification more specifically and how H2S can regulate it.

 Rebuttal: Very professional question. Indeed, microcalcification of plaque is essential in clinical imaging diagnosis. Many clinical trials demonstrated microcalcification positively correlating with acute coronary syndrome. In mechanism, large calcification and microcalcification share the similar process and modulation.

  • A paragraph should be added to clarify the potential role of H2S on plaque stability and the prevention of cardiovascular events. Indeed, much of the literature cited in this manuscript concerns animal models and the translation to humans remains to be established. The authors have discussed this point in the conclusion part, but it would be more important to discuss it in a separate paragraph.

 Rebuttal: Thank your sincere advance. We rewrote the paragraph on conclusion and perspective. We discussed some of the important issues, possible solutions, and potent treatment of H2S donors.

Round 2

Reviewer 1 Report

No further comments.

Reviewer 2 Report

I checked and I’m satisfied with this revised version of manuscript. Thanks to this author’s great effort, I think this manuscript holds enough value to be published.

Reviewer 3 Report

No further comments